# Succinct Explanations with Cascading Decision Trees

## Abstract

Classic decision tree learning is a binary classification algorithm that constructs models with first-class transparency - every classification has a directly derivable explanation. However, learning decision trees on modern datasets generates large trees, which in turn generate decision paths of excessive depth, obscuring the explanation of classifications. To improve the comprehensibility of classifications, we propose a new decision tree model that we call Cascading Decision Trees. Cascading Decision Trees shorten the size of explanations of classifications, without sacrificing model performance overall. Our key insight is to separate the notion of a decision path and an explanation path. Utilizing this insight, instead of having one monolithic decision tree, we build several smaller decision subtrees and cascade them in sequence. Our cascading decision subtrees are designed to specifically target explanations for positive classifications. This way each subtree identifies the smallest set of features that can classify as many positive samples as possible, without misclassifying any negative samples. Applying cascading decision trees to new samples results in a significantly shorter and succinct explanation, if one of the subtrees detects a positive classification. In that case, we immediately stop and report the decision path of only the current subtree to the user as an explanation for the classification. We evaluate our algorithm on standard datasets, as well as new real-world applications and find that our model shortens the explanation depth by over 40.8% for positive classifications compared to the classic decision tree model.

## 1 Introduction

Binary classification is the process of classifying the given input set into two classes based on some classification criteria. Binary classification is widely used in everyday life: for example, a typical application for binary classification is determining whether a patient has some disease by analyzing their comprehensive medical record. Existing work on binary classification mainly uses the accuracy of prediction as the main criterion for evaluating model performance. However, in order for a model to be useful in real-world applications, it is imperative that users are able to understand and explain the logic underlying model predictions. Model comprehensibility[1] in some real-world applications, especially in the medical and scientific domains, is of the utmost importance. In these cases, users need to understand the classification model to scientifically explain the reasons behind the classification or even rely on the model itself to discover the possible solution to the target problem.

It is difficult to provide explainability without sacrificing classification accuracy using current models. "Black-box" models such as deep neural network, random forests, and ensembles of classifiers tend to have the highest accuracy in binary classification Freitas (2014); Doilovi et al. (2018). However, their opaque structure hinders understandability, making the logic behind the predictions difficult to trace. This lack of transparency may further discourage users from using the model Augasta & Kathirvalavakumar (2012); Van Assche & Blockeel (2007).

Decision tree models, on the other hand, have transparent decision making steps. A traversal of features on the decision path from the root to the leaf node is presented to users as a rule. Therefore, compared to other models, the decision tree model has historically been characterized as having

---

[1]In this paper, comprehensibility and interpretability are used interchangeably.

high comprehensibility Freitas (2014); Doilovi et al. (2018). However, whether models generated by classic decision trees provide enough comprehensibility has been challenged: "decision trees [...] can grow so large that no human can understand them or verify their correctness" Caruana et al. (1999), or they may contain subtrees with redundant attribute conditions, resulting in potential misinterpretation of the root cause for model predictions Freitas (2014).

The work presented in this paper introduces an algorithm for deriving succinct explanations for positive classifications, while maintaining the overall prediction accuracy. To this end, we introduce a novel *cascading decision tree* model. A cascading decision tree is a sequence of several smaller decision trees (subtrees) with the predefined tree depth. After every subtree sequentially follows another subtree, mimicking in this way a cascading waterfall, thus the name. The sequence ends when the subtree does not contain any leaves describing positively classified samples. Fig. 2 depicts one such cascading decision tree.

The main idea behind cascading decision trees is that, while most algorithms for constructing decision trees are greedy and they try to classify as many samples as soon as possible, such classification results in large explanation paths for the samples in the lower levels of the tree. Instead, we construct a subtree of the predefined depth. That subtree contains a short explanation for the samples it managed to classify. However, the subtree with a short depth will misclassify samples. We next repeat the process on the training set with the samples that were previously classified positively removed. This way, the samples classified as positive in the second subtree will have a much shorter explanation path than they would in the original decision tree. In the cascading decision tree model, an explanation path for positively classified sample is the path that starts in the root of the corresponding subtree.

We target explanations for only positive classifications, based on real-world motivation. In the medical domain, a positive classification result indicates that a person has the disease for which the test is being done NIH (2020). The positive classification is also combined with additional testings needed for a full diagnosis CDC (2020). Note that, if a practical application arises, our cascading decision trees model could easily be changed to target the negative classifications.

Reducing the size and the depth of a decision tree to improve comprehensibility has been studied, both from a theoretical and a practical perspective. However, constructing such optimally small decision trees is an NP-complete problem Hyafil & Rivest (1976), and the main drawback of these approaches is that the model is computationally too expensive to train. Even when using the state-of-the-art libraries Aglin et al. (2020); Verwer & Zhang (2019), we observed that complexity. To illustrate this on an example, to learn a model on the Ionosphere dataset (from the UCI Machine Learning Repository), the BinOCT tool needs approximately 10 minutes, while our approach completes this task in 1.1 seconds.

We demonstrate the applicability of the cascading decision tree model in two ways. First, we use our model to perform standard binary classification on three numerical datasets from the UCI Machine Learning Repository. Second, we apply our model to a new application of binary classification, namely continuous integration (CI) build status prediction Santolucito et al. (2018). Overall, we report that compared to the classical decision tree algorithm, our approach shortens the explanation depth for positive classifications by more than 40.8% while maintaining the prediction accuracy.

## 2 MOTIVATING EXAMPLES

In this section we demonstrate how cascading decision trees can generate a shorter and more succinct explanation.

The following simple synthetic example is contrived to illustrate our tool's basic functionality. Given the dataset in Table1, a classical decision tree will construct a model shown in Fig. 1. Using the same dataset, our cascading decision trees algorithm generates a model with three subtrees in shown in Fig. 2. Let's assume that there is a new sample ,"Sample11", with the feature vector $(F, F, F, T)$. Both models classify "Sample11" with the same prediction result, "Positive". However, the explanations extracted from these two models are different.

In the classic decision tree model (Fig. 1), "Sample11" falls into node $(9)$. Thus, the explanation path here is "Feature1 = F", "Feature2 = F" and "Feature3 = F", with the explanation depth of three.

Table 1: Synthetic Dataset for Binary Classification.

|  | **Feature1** | **Feature2** | **Feature3** | **Feature4** | **Label** |
|---|---|---|---|---|---|
| Sample1 | T | T | T | F | Positive |
| Sample2 | T | T | F | F | Positive |
| Sample3 | T | T | T | F | Positive |
| Sample4 | T | T | T | F | Positive |
| Sample5 | F | F | F | T | Positive |
| Sample6 | F | T | F | F | Positive |
| Sample7 | T | F | F | F | Negative |
| Sample8 | F | T | T | F | Negative |
| Sample9 | F | F | T | F | Negative |
| Sample10 | F | T | F | F | Negative |

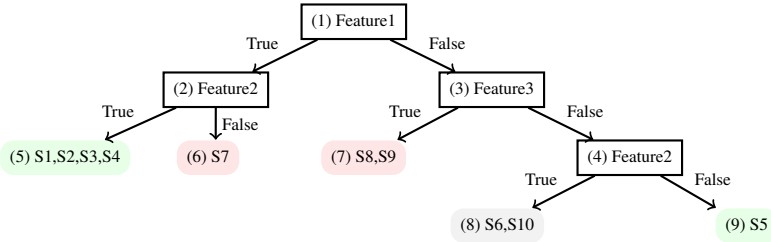

Figure 1: An example of a full classic decision tree on dataset given in Table 1 with green boxes as Positive nodes, red boxes as Negative nodes, and gray boxes as Mixed nodes.

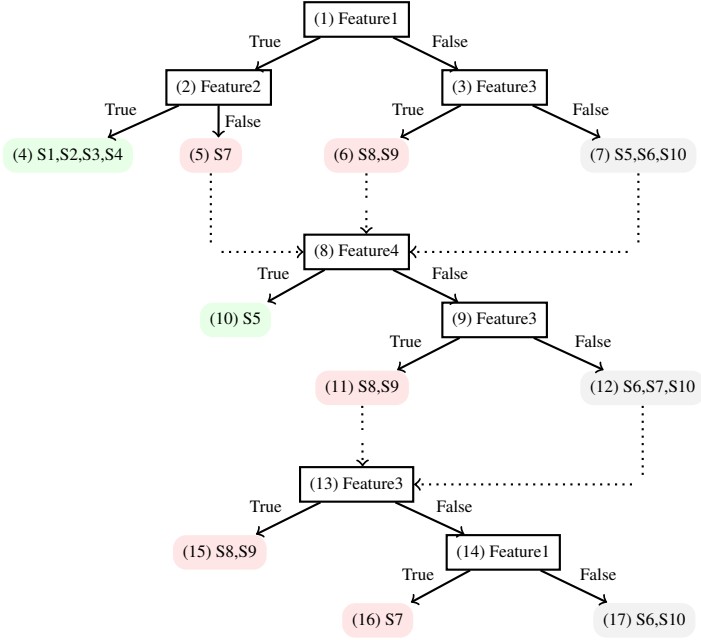

Figure 2: An example of Cascading Decision Trees on dataset given in Table1, with maximum depth of three in each subtree.

Using the cascading decision tree model (Fig. 2), "Sample11" first falls into node (7) in the first subtree. Node (7) is not classifying positive samples, so "Sample11" is passed to the second subtree and eventually falls into node (10). This way, the explanation path for"Sample11" is "Feature4 = T", with an explanation depth of only one.

## 3 RELATED WORK

**Ensemble Methods.** The cascading decision trees algorithm pursues a different goal than existing ensemble methods, such as Bagging Breiman (1996) and Boosting Schapire (1990). Ensemble methods for decision trees combine multiple weaker classifiers into a larger classifier with the goal of increasing overall accuracy, while cascading decision trees focus on shortening explanations. Bagging divides the dataset into multiple, smaller datasets, generates the classifier for each dataset and produces the final classification via a voting process. Boosting algorithms iteratively learn weak classifiers, gives them weights, and adds them one after another to generate a final strong classifier. The combination strategies of these ensemble methods obfuscate the traceability of a classic decision tree, which is the source of simple explanations for classifications Zhao & Sudha Ram (2004). The limited model comprehensibility of Bagging and Boosting is also shared by other ensemble methods, which makes their output difficult to interpret.

**Oblique Decision Trees.** Oblique decision trees Murthy et al. (1994) extend the classic decision tree model by allowing each decision node to combine checks against multiple features. This series of work contains a rich family of models such as multivariate decision trees Brodley & Utgoff (1995), loose and tight coupling Gama & Brazdil (2000), and constrained cascade generalization of decision trees Zhao & Sudha Ram (2004). However, compared to cascading decision trees, this oblique design is not only the training stage computationally expensive Lee & Jaakkola (2020), but also makes the explaining a classification in the decision-making stage opaque and hard to interpret Zhao & Sudha Ram (2004).

**Optimal Decision Trees.** Optimal decision tree algorithms learn models with the optimal prediction accuracy under the constraints of the predefined depth of the tree. However, learning an fixed depth optimal decision tree without heuristics is an NP-complete problem Hyafil & Rivest (1976). The main drawback of using this approach in practice is that the model is too computationally expensive to train. For example, the state-of-the-art optimal decision trees implementations Aglin et al. (2020); Verwer & Zhang (2019) need ten minutes to train the ionosphere dataset[2]. On the same dataset, we show in our evaluation (Sec. 5) that our cascading decision trees learning process terminated in seconds with competitive accuracy (cf. Verwer & Zhang (2019)). In addition, instead of being a direct competitor, an optimal decision tree learner could be used as the base decision tree inducer in our algorithm to potentially improve the prediction accuracy, though at the cost of training time.

**Model Interpretability.** The comprehensibility of classification models, including decision tree models, has been extensively explored Doilovi et al. (2018). Specifically, Quinlan (1987) demonstrated that small models are more interpretable than larger ones in the decision tree. Similarly, Huysmans et al. (2011) ran a user study to illustrate that larger representations result in a decrease in both the accuracy of user's answers and users confidence in the model itself. This work supports the motivation that minimizing explanations is a valuable direction to explore.

## 4 METHODOLOGY

**Formalism.** For interpretability, we take inspiration from the LIME system Ribeiro et al. (2016), which states that interpretable explanations should "provide qualitative understanding between joint values of input variables and the resulting predicted response value". In our work, we specifically focus on binary classification, and only target explanations for positive classifications. An explanation in our system should capture the essence of the positive classification. That is, the explanation identifies the key input values that contributed to the positive classification. The classification should still be the same, no matter the state of the input values not included in the explanation. It is possible the sample will be classified positively in a different way, but as long as the input values in the explanation are unchanged, the classification will still be positive.

To define a valid explanation, we consider a dataset of samples with binary labels. Without loss of generality (as we consider only the decision tree setting), let $(x, y) : (2^n, C)$ denote a sample as an $n$-dimensional Boolean vector and its ground-truth classification respectively. The number of classes is $|C|$ - in our application we consider only binary classification, where $|C| = 2$, though

---

[2]https://archive.ics.uci.edu/ml/datasets/Ionosphere

the definition of valid explanations generalizes to larger $|C|$. Given a test set $T : \{(x, y)\}$, we can learn a model $M : 2^n \to C$ which maps input vectors of samples to a classification. An explanatory model, $M^+ : 2^n \to \mathcal{E}$ expands upon the model $M$ to additionally map input vectors to explanations, $\mathcal{E}$. An explantation $\mathcal{E} : 2^n$ is a Boolean vector, where every true element means that the feature at the corresponding index in the input vector is a necessary part of the classification result. The size of an explanation, $|\mathcal{E}| = \sum_{i \in n} \mathcal{E}[i]$, is the number of true elements in the explanation vector.

We define the $validity$ of an explanation via array multiplication. An explanation, $\mathcal{E}$, is valid over an input vector $x$, when any part of the input vector not captured by the explanation may be modified without changing the classification result of the model.

$$isValid(\mathcal{E}, x) \iff \forall z \in 2^n.\ M(x) = M(\mathcal{E} * x + z * \neg \mathcal{E})$$

**Example 1** A trivial explanation is always available for any sample - the explanation that every element of the sample vector is important. From Fig. 1, for input vector "Sample5", a valid explanation is $\mathcal{E} = [1, 1, 1, 1]$. This means that as long as no features change, the classification remains the same.

**Example 2** In general, a shorter explanation can be extracted from classic decision trees. As an example, the explanation of "Sample5" from the decision tree in Fig. 1 is the explanation that keeps only the elements in the decision path of the tree: $\mathcal{E} = [1, 1, 1, 0]$. This is the commonly accepted approach to explanations of decision tree classifications.

**Example 3** An even shorter explanation is possible with cascading decision trees. Looking again to "Sample5" as classified by the cascading decision tree in Fig. 2, a valid explanation keeps only "Feature 4" - $\mathcal{E} = [0, 0, 0, 1]$. Although changes to "Feature 1-3" may change the decision path, the classification remains the same as long as "Feature 4" is unchanged.

A explanatory model is valid if and only if the provided explanation is valid for all possible inputs, that is: $\forall x \in 2^n.\ isValid(M^+(x), x)$. We note that a valid explanatory model only needs to correctly *explain* its classifications for all inputs - not correctly classify all inputs.

The above definition of a valid explanation is always true of the decision path of a classic decision tree. This aligns with the notion that the decision path of a classification in a classic decision tree is a valid explanation for the resultant classification of a sample. As we will show, this definition of a valid explanation is also always true of the shortened "explanation path" we propose with our cascading decision trees. Cascading decision trees provide explanations only over samples classified as positive, though the definition is more general.

**Building Cascading Decision Trees.** The cascading decision trees algorithm is described through pseudocode in Algorithm 1. Our insight is in the training process - it is the goal of each cascading decision subtree to identify the smallest set of features that can classify as many `Positive` samples as possible, without misclassifying *any* `Negative` samples. Specifically, the procedure Fit builds a classic decision tree, `clf`, on our training set, $\mathcal{TR}$ and adds it to the cascading tree list - initially this list is empty. For every leaf node we compute the `mixed` value, which is the percentage of samples from $\mathcal{TR}$ classified by this leaf node that are also `Positive` samples. `Positive` nodes are leaf nodes with a `mixed` value greater than the threshold (we use 80% in this paper). If `clf` has no `Positive` nodes, it means we have learned a sufficiently good classifier and we stop (Line 11). Otherwise, we first remove samples truly classified by `Positive` nodes in this `clf` from $\mathcal{TR}$, and then obtain a new $\mathcal{TR}$ to use in the next iteration of the loop (Line 14).

Our cascading decision trees algorithm1 presented here has no pruning phase. However, our algorithm is generic enough to be combined with any pruning techniques Esposito et al. (1997) and different goodness measurement for decision nodes split, such as entropy Shannon (1948) and gini impurity Havrda et al. (1967).

The time complexity of cascading decision tree algorithm is bounded by the size of the training set. Suppose the training time for building classic decision trees is a function of the number of the training samples, $n$ and the number of features $k$. We use the decision tree module in scikit-learn[3] as our base classic decision tree inducer, which is built upon the CART Breiman et al. (1984) method. Since features are recursively reused in every decision node based on a numerical splitting criterion,

---

[3]https://scikit-learn.org/stable/

---

**Algorithm 1** Build Cascading Decision Trees Classifier

---

    **Input:** $\mathcal{TR}$ :Labelled Training Set
    **Input:** $\theta$ :Threshold
    **Input:** $depth$ :Maximum depth of each decision subtree
    **Output:** $cascadingTree$ :A list of cascading decision subtrees
 1: **procedure** FIT($\mathcal{TR}, \theta, depth$)
 2:    $done$ = False
 3:    $cascadingTree$ = []
 4:    **while** $done = False$ **do**
 5:        $clf$ = classicDecisionTree.fit($\mathcal{TR}, depth$)
 6:        $cascadingTree$.append($clf$)
 7:        $done$ = True
 8:        **for** $leafNode$ **in** $clf$ **do**
 9:            **if** $(mixed(leafNode) >= \theta)$ **then**
10:               $done$ = False
11:        **if** $(done = True)$ **then**
12:            **Break**
13:        **else**
14:            $\mathcal{TR}$ = filter ($clf$.predict($sample, \theta$) = Positive, $\mathcal{TR}$)
    **return** $cascadingTree$

---

the depth of the decision tree is bounded by the number of the training samples $n$ in our model. Therefore, the time complexity for building one base decision tree is bounded by $\mathcal{O}(n^2 k)$.

According to our cascading decision trees algorithm1, after building one decision subtree, samples that are classified as `True Positive` are removed from the next round of decision subtree construction. In the worst case, every time, only one `True Positive` is classified in the current decision tree. The time complexity for building the next cascading decision subtree is in $\mathcal{O}((n-1)^2 k)$. Therefore, the overall cascading decision trees training time $T$ is bounded by: $\mathcal{O}(\sum_{m=1}^{n} m^2 k) = \mathcal{O}(\frac{n(n+1)(2n+1)}{6}k) = \mathcal{O}(n^3 k)$.

**Testing Cascading Decision Trees.** Our cascading decision trees testing process is described in Algorithm 2. In the procedure Test, we run all decision trees sequentially, and we report the decision path of that tree only as our explanation path.

---

**Algorithm 2** Test Cascading Decision Trees Classifier

---

    **Input:** $cascadingTree$ :A list of cascading decision tree classfier
    **Input:** $\theta$ :Threshold
    **Output:** Prediction Result: A boolean variable, Postive or Negative.
    **Output:** Explanation Paths: A conjunction of boolean statements to explain the decisions.
 1: **procedure** TEST($cascadingTree, \theta$)
 2:    **for** $clf$ **in** $cascadingTree$ **do**
 3:        **if** $clf.predict$(sample, $\theta$) $= Positive$ **then**
 4:            **return** (Postive, $clf$.path(sample))
    **return** (Negative)

---

## 5   EVALUATION

This section aims to evaluate the cascading decision trees algorithm by answering the following questions: (1) compared to the classic decision trees algorithm, what is the percentage of explanation depth for positive classifications that has been shortened by using cascading decision trees? (2) what is the prediction accuracy and the turn-around efficiency of cascading decision trees algorithm? (3) how well does cascading decision trees algorithm perform in real-world edge application such as continuous integration (CI) build status prediction? All experiments were conducted on a MacBook Pro with a 2.5 GHz Intel i7 processor with 16GB of RAM.

### 5.1 EMPIRICAL EVALUATION

To empirically evaluate the explainability and accuracy of the cascading decision trees algorithm, we collect three datasets from UCI machine learning repositories[4] as our benchmark set. The UCI machine learning datasets are standard benchmarks for comparing the performance of tree-based classification methods. We select these three binary classification datasets out of the 32 UCI datasets Zhao & Sudha Ram (2004). We select only the binary classification tasks, as our method provides explanations for positive classifications. While our method could be applied to multiclass classification, the user must pick a class to focus on creating succinct explanations, which is not clear from the UCI datasets. Additionally, we select only the datasets with exclusively numeric features, as the underlying library, sccikit-learn[5], of our implementation is limited in this respect.

Each of the three UCI datasets reflect tasks clearly benefit from more succinct explanations for positive predictions. They are:

1. Classification of breast cancer. "Positive" samples mean the tumor turns out to be malignant, while "Negative" samples are benign.

2. Detection of free electrons in the ionosphere. "Positive" samples show the detecting signals fail to detect the free electrons; the signals just pass through the ionosphere. "Negative" samples show some evidence of the stucture in the ionosphere.

3. Discrimination of objects. This dataset includes bouncing sonar signals off a mine (metal cylinder) at various angles and under various conditions. "Positive" samples indicate the object is indeed a rock not a mine, while "Negative" samples indicate the object is a mine.

We compare the performance of the proposed cascading decision trees algorithm to the performance of the classic decision trees algorithm with various bounds on the depth of the learned tree. We quantify algorithm performance using five-fold cross validation, and randomly shuffle the datasets. The maximum depth of the cascading decision trees algorithm is uniformly set to three in all tests. The threshold in cascading decision trees algorithm is set to a fixed number 0.8.

**Explainability.** The shorter an explanation, the more comprehensible that explanation is to users Quinlan (1987); Huysmans et al. (2011); Pazzani (2000). The ability to clearly explain the output of machine learning models is critical to their acceptance and use in practice, an idea summarize by Huysmans et al. (2011) - "Larger representations result in a decrease in user's answer accuracy and a decrease in user's confidence to the model."

Table 2: Average Explanation Depth of Postive Classifications from Cascading Decision Trees and Classical Decision Trees.

| Dataset | Classical Decision Trees | Cascading Decision Trees | Improvement |
|---------|--------------------------|--------------------------|-------------|
| Breast cancer | 2.658 | 1.991 | 25.1% |
| Ionosphere | 2.694 | 1.418 | 47.4% |
| Sonar | 3.813 | 1.943 | 49.0% |

Table 2 shows the comparison of the average explanation depth of model generated by cascading decision trees and classic decision trees. Our cascading decision trees algorithm shortens the explainable paths to users by 40.8% on average among three datasets. The average representation size of our cascading decision trees model is 1.78 among three datasets. We specifically focus on the explanations for positive classifications. This means on average, only 1.78 features are necessary for the classification of a positive sample in cascading decision trees. However, for classical decision trees, 3.06 features are necessary for the classification of a positive sample. This succinctness of the explanation of positive predictions enhances qualitative understanding to users, which could be used to diagnose the medical causals and back up the scientific hypothesis.

**Accuracy.** Table 3 shows the breakdown empirical evaluation of cascading decision trees. In two out of three datasets, our cascading decision tree surprisingly outperforms the base classic decision trees in prediction accuracy.

---

[4]http://archive.ics.uci.edu/ml/index.php/

[5]The decision trees module in scikit-learn is also used as the implementation for the classic decision trees algorithm in our evaluation.

Table 3: Breakdown Evaluation of Cascading Decision Trees on Three Classic UCI datasets.

| Dataset | Cascading | Explanation Depth | Accuracy | Runtime (s) | TP | TN | FP | FN | Precision | Recall | F-1 Score |
|---|---|---|---|---|---|---|---|---|---|---|---|
| Breast cancer | On | **1.991** | 93.51% | 1.039 | 36.8 | 69.8 | 1.8 | 5.6 | 95.42% | 86.57% | 90.54% |
| | Off | 2.658 | 93.16% | 0.068 | 38.4 | 67.8 | 3.8 | 4.0 | 91.38% | 90.47% | 90.80% |
| | Off (max_depth = 3) | 2.311 | 91.40% | 0.062 | 34.8 | 69.4 | 2.2 | 7.6 | 93.88% | 82.07% | 87.53% |
| Ionosphere | On | **1.418** | 88.73% | 1.108 | 20.4 | 42.6 | 3.0 | 5.0 | 88.28% | 80.89% | 84.37% |
| | Off | 2.694 | 85.92% | 0.060 | 20.0 | 41.0 | 4.6 | 5.4 | 81.77% | 80.16% | 80.68% |
| | Off (max_depth = 3) | 1.483 | 84.23% | 0.051 | 15.8 | 44.0 | 1.6 | 9.6 | 92.42% | 61.57% | 73.60% |
| Sonar | On | **1.943** | 66.19% | 0.467 | 12.2 | 15.6 | 4.2 | 10.0 | 78.63% | 55.92% | 62.46% |
| | Off | 3.813 | 74.29% | 0.055 | 17.4 | 13.8 | 6.0 | 4.8 | 74.45% | 78.29% | 76.06% |
| | Off (max_depth = 3) | 2.658 | 65.71% | 0.049 | 11.0 | 16.6 | 3.2 | 11.2 | 79.75% | 50.42% | 60.32% |

To lower the explanation path of the decision trees model, one common technique is to set the maximum depth to a classic decision trees classifier. However, compared with the cascading decision trees, setting the maximum depth to three incurs a decrease in average prediction accuracy by around 4.0%. Even with this max depth and lower accuracy, in all three datasets, the average explanation path is still longer than cascading decision trees algorithm.

In conclusion, compared with the classical decision trees model both with and without a fixed depth, our cascading decision trees algorithm delivers better model comprehensibility via significantly shorter explanation depth while maintaining high prediction accuracy.

**Low False-positive Rate.** In addition to better model comprehensibility, the cascading decision trees model has another key advantage, which is the low false-positive rate in prediction. In medical and scientific domains, explanations behind positive classification results are often more useful than explanations behind the negatives. For example, in the medical domain if the prediction result is positive, the doctor needs to justify the accurate reasons behind the diagnosis of the disease and then report them to patients. A high false-positive rate not only greatly reduces physicians' confidence in adopting the prediction result but also leads to unnecessary and invasive follow-up tests on patients HNR (2020). Therefore, it is crucial to have a competitive accuracy prediction model with very low false-positive rate. As shown in Table 3, the cascading decision trees algorithm has the lowest false positive (FP) rate for all three datasets compared to the classical decision trees algorithm.

**Turn-around Efficiency.** Although this paper does not focus on building fast classifier, the turn-around efficiency of our cascading decision trees algorithm turns out to be great in practice. The training process finishes in seconds in all three real-world UCI datasets. Taking the ionosphere dataset as an example, our cascading decision trees learning process terminates in 1.1 seconds with accuracy of 88.7%. This is comparable to accuracy of 87.0% of the state-of-the-art optimal decision trees algorithm BioOCT when depth of the tree are three (cf. Verwer & Zhang (2019)). However, BioOCT takes around ten minutes to train.

**Real-world Application.** The second part of our evaluation was to apply the cascading decision trees for predicting the build status in the continuous integration (CI) environment Gousios et al. (2014). Notifying that the attempt to build the project might fail, could save software engineers countless hours. We ran a study (link omitted for anonymity) where we learn how to classify potential build failures. The evaluation results demonstrate that cascading decision trees provides a shorter explanation, with a competitive prediction accuracy of 90.55%. Moreover, the ratio of the average number of false positives reports to the average number of correct classifications is only 1.8% . Our study shows that the use of cascading decision trees provides developers with a more succinct but comprehensible set of rules that are responsible for positive classifications.

## 6 CONCLUSIONS

Learning decision trees on modern datasets generates large trees, which in turn produce decision paths of excessive depth, obscuring the explanation of classifications. This paper intends to maximize model comprehensibility while maintaining prediction accuracy in binary classification. The cascading decision trees algorithm has been proposed to provide more succinct explanations in binary decision trees. We evaluated our algorithm in real-world medical, scientific and program analysis datasets, where the explainability of the positive test result is of the utmost importance. Our cascading decision trees algorithm shortens the explanation depth by over 40.8% for positive classifications compared to the classic decision trees algorithm.

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

## A    APPENDIX

**Real-world Application.** Encouraged by the prior results from our empirical study, we conduct our second part of our evaluation by using our cascading decision trees algorithm in a novel real-world application, continuous integration (CI) build status prediction.

In CI environment, developers upload their code as one commit, and CI starts to build and tests users' code when they finish their uploading. Feedback from the positive CI build status enables developers to quickly locate and troubleshoot errors in the code, expediting the software development cycle. However, one major drawback of CI is that a CI build attempt can be extremely time consuming, and can even take longer than a day Santolucito et al. (2018), discouraging the usage of CI itself. Previous work Santolucito et al. (2018) used the historical data stored in the CI environment to predict the CI build status and report the root causes behind the failures to users.

We choose this dataset because it is an ideal testbed for our cascading decision trees algorithm, binary classification task on all numerical features. As opposed to other common source of programming error such as buffer overflow, divide by zero and so on Wang et al. (2013), CI build failure is mainly caused by library version inconsistency. Therefore, all features extracted in this dataset are numerical, and as a result, is amenable to analysis by our algorithm. Moreover, in program analysis domain, a very low false-positive rate is crucial for user acceptance of the tool Junker et al. (2012).

Table 4: Breakdown Evaluation of Cascading Decision Trees on CI Build Status Prediction.

| Cascading | Explanation Depth | Accuracy | Runtime (s) | TP | TN | FP | FN | Precision | Recall | F-1 Score |
|---|---|---|---|---|---|---|---|---|---|---|
| ON | 2.18 | 90.55% | 249.43 | 68 | 95 | 135 | 1428 | 66.50% | 58.70% | 62.36% |
| OFF | 2.31 | 90.50% | 277.95 | 69 | 95 | 135 | 1427 | 66.18% | 58.70% | 62.21% |

Table 4 shows the breakdown evaluation of using cascading decision trees on CI build status prediction. The strategy we employ in the evaluation is the following; for a repository with $n$ total historical commits, we build our model with $n/2$ commits, and use that model to evaluate commit $(n/2) + 1$. We then rebuild the model with a training set of $(n/2) + 1$ commits and used that incremental model to predict the status of the next coming commit. Using this approach, we find that cascading decision tree classifier can predict build status with an overall accuracy of 90.55%, which is almost the same as the accuracy of 90.50% if using only classic decision tree classifier. However, in addition to competitive prediction accuracy, cascading decision tree shortens the explanation depth for failed builds by 5.6%. This could help developers identify error locations more quickly and shorten the software development cycle.

In addition, our evaluation results show that the ratio of the average number of false positive reports to the average number of correct classifications is only 1.8% (FP/TP+TN). In conclusion, the use of cascading decision trees provides developers with a more succinct but comprehensible set of rules that are responsible for positive classifications. This allows developers to more accurately and confidently troubleshoot the CI build failure.

