# OpenReview forum: "Succinct Explanations with Cascading Decision Trees"
_ICLR.cc/2021/Conference — Reject_

### Official Review · AnonReviewer3 · 2020-10-23
**A novel idea that needs to be further polished.**

**Rating:** 4
**Confidence:** 5

**Review:**

The authors presented in this submission a nice novel idea of building a tree ensemble in a cascading style so that any positive predictions are decided and explained by the first tree predicting them positively. The reviewer finds this idea very interesting and clearly elaborated in this paper. However, more theoretical and empirical justification is crucially necessary in order to make the claims in the submission convincing. The issues listed here are some questions that the reviewer believes should have been discussed or answered in the paper.

Major issues:

Despite the fact that the focus of cascading decision trees (CDTs) is a short explanatory path, my major concern here is that it is very hard to justify them as a proper statistical model.

1.
Practically,  there are very simple adversarial cases that the cascading decision trees will fail to build. Consider this example that X ~ Unif([0,1]^2) and Y|X == 1 if X \in [1/3, 2/3]^2 otherwise 0. If we apply the algorithm in the paper, using depth=2 trees and a mixed node threshold theta = 0.5 (which I believe are a reasonable choice), the positive-focused CDTs will almost fail to construct the first tree with a decently large sample, let alone the cascading subsequence. On the other hand, since depth=2 trees can cover all rectangles (0,0) - (a,b) \in R^2,  the counterpart random forests or GBDTs are universal approximators.

By also checking the negative predictions there might be ways to mitigate this issue. However relevant discussions are lacking in this paper in its current shape.

From a more abstract perspective regarding the CDTs depth: in a d-dim sample space, to separate out a d-dim cube, we are likely in the need of 2d splits. There might be fewer splits needed when the cube is touching the boundary of the support or there are other nodes made before as in an ensemble. But in general we should not make such assumptions, and it would be better if we could have more discussions in the paper.

2
Classic classification trees are asymptotically bayesian classifiers, whereas we can imagine asymptotically CDTs assign the positive label to a region only when the true positive rate theta* within the region is larger than the constant threshold theta, which means CDTs are intrinsically inaccurate - more specifically, low recall as mentioned by the authors. This situation is further worsened as CDTs will take positive examples off from the sample. The fact that CDTs are theoretically incapable of finding all positive examples is harming their credibility of giving short explanations to positive predictions. A bandage here might be to use adaptive thresholds, but no discussions are currently present in the submission.

Minor issues:

1. Classic CARTs are very sensitive towards outliers. The "split one example out each time" scenario being analyzed in the paper is likely to cause overfitting, chasing the outliers, and instability. It should be avoided.
2. CARTs default greedy building algorithm uses entropy or Gini index which are indifferent towards both positive and negative examples. Since the focus in the paper is on positive predictions, it is worth discussing how the algorithm should be changed accordingly.
3. It would be better if there were instructions in the paper regarding choosing the mixed node threshold theta, the tree depth, and hopefully the ensemble size.
4. Since CDTs are still a tree ensemble, their capacity is expected to be larger than a single decision tree which can even be a bit deeper. The results pertaining to the accuracy, precision and recall in the empirical study session are therefore slightly unfair comparison - it would be better to benchmark against decision tree ensembles or rule-based models [1].

[1] Wang, Fulton, and Cynthia Rudin. "Falling rule lists." Artificial Intelligence and Statistics. 2015.

---

### Official Review · AnonReviewer4 · 2020-10-27
**An interesting twist of decision trees that shortens the paths for improved explainability, but there are some concerns with the proposed technique**

**Rating:** 3
**Confidence:** 4

**Review:**

This work proposes to use cascade decision tree models to come up with shorter explanations for the predictions made by the decision trees.

The proposed technique focuses on explaining one class in a binary classification task, i.e., positive samples. The idea is to build a decision tree with predefined depth to classify positive samples, remove those classified positive, as well as negative samples in leaf nodes that are dominated by negative samples, from the dataset, and then repeat this process until the sequence of tree models is built.

Explainability of ML models is an important topic, especially for medical scenarios. In addition, shortening the decision tree paths to improve explainability is a promising direction. The writing of the work is also clear and easy to follow.

Having said that, I have the following concerns about this work:

W1. The application scenario is narrow. And it is not clear why the explanation path is not the concatenation of all the paths of the cascading trees.

D1. The proposed approach only applies to binary classification task. It is not clear how this can extend to multi-class and regression models.

D2. Also, even for binary classification, it only explains one class. It is counterintuitive that you need to build different models to explain the classification of the two classes from the same dataset.

D3. Since the process of building a subtree is independent of the previous subtrees built, i.e., the features and splits of the tree being built does not consider the features and splits in previous subtrees, it seems to me that the subtree is pre-conditioned by the previous subtrees. It is not clear why it is correct to only use the subtree that the prediction ends for explanation instead of all the subtrees that are used before the prediction ends.

W2. The complexity of this decision tree can be high, leading to high inference time. And the model can be overfit to positive samples.

D4. Because the cascading model tries to construct leaves with most pure positive samples, the total depth and the number of trees in the model can be quite high. This will especially impact the overhead in the inference.

D5. Also, given this tree is built for optimizing the classification of one class with very high accuracy, the model can be overfit to that class, and the prediction accuracy of the other class is not guaranteed.

W3. The evaluation needs to be enhanced.

D6. The average path length of classic decision trees is already small, i.e., less than 4. The evaluation needs to perform on more complex datasets.

D7. In medical scenarios, we would expect the negative data samples are much more than positive data samples. It is already easy to overfit for the positive samples in this scenario, and as mentioned in D5, the technique itself also tends to overfit. Overfitting may not be a good idea for this skewed dataset.

D8. It is unclear if the proposed technique gives better explanation without doing a user study. As mentioned in W1 and D3, it is not entirely clear why the explanation does not need to concatenate all the paths in the subtrees before the prediction ends. This is especially a concern since the classic decision tree only has a single tree. It will be great to conduct some user study to understand why the proposed technique impacts the Explainability.

---

### Official Review · AnonReviewer2 · 2020-10-28
**Interesting idea with a too limited empirical evaluation**

**Rating:** 5
**Confidence:** 3

**Review:**

*Summary*
This paper introduces the Cascading Decision Tree, a novel variant of decision trees with permits to extract short explanations for a class of interest. The idea is to realize a cascade of small decision trees: at a certain level, the tree is built using all points except the positive ones correctly classified by trees in previous levels. The method has been tested using three standard datasets and a novel application.

*Positive points*
-> The motivation of this paper is definitely interesting. Actually, interpretability represents one of the key features of decision trees: deeper trees, which may be needed for solving complex classification tasks, may loose this property.
-> The proposed approach is simple but reasonable.
-> The paper is well written and clear.



*Negative points/questions*

-> The main problem of the proposed method is the empirical evaluation:
i) authors used only three ML datasets of moderate size
ii) the impact of different parameters of the proposed methods is not analysed. For example, which is the impact of the threshold? (Authors only used 0.8 in all experiments). Further, all trees in the cascading architecture have max depth of 3. What about using longer/shorter trees?
iii) Comparisons of results in tables are not supported by a statistical test. Is the accuracy of 93.51% (first row table 3) different than 93.16% (second row)? Without a statistical test it is impossible to derive significant conclusions. A possible strategy: shown results are computed using 5 fold cross validation, thus meaning that reported numbers represent the average of 5 repetitions; by repeating the whole procedure 10 times (with different random subdivisions of training and testing), author would have a more robust estimation of the reported numbers (average of 50 tests). Moreover, with such scheme they can also assess the statistical significance of differences by using a (paired) t-test.
iv) the explanation depths of classic trees are in average very short, can authors comment this? Moreover,  which is the maximum length of the classic decision trees? Authors simply reported that they used “various bounds on the depth of the learned tree”.
v) conclusions derived from tables are not completely supported by the tables. For example, in the paragraph “Low False-positive Rate” authors say that “As shown in Table 3, the cascading decision trees algorithm has the lowest false positive (FP) rate for all three datasets compared to the classical decision trees algorithm.”. By looking at table 3 this is not evident: it seems to me that with Ionosphere and Sonar the lowest FP is obtained with the variant without cascading (“Off (max_depth =3)”). Am I correct?
vi) I wonder if the comparison with BioOCT is fair (in terms of classification accuracy), since the maximum depth of this last method has been fixed to three.
vii) Comparisons only involve standard trees: what about comparing also to other methods used to reduce the depth of decision trees? This would enlarge the scope of the experiments and the value of the proposed approach.


-> I think that the proposed approach can be better inserted into the state of the art. Actually, the combination of different classifiers in cascade is not new in the literature of ensemble classifiers, with many methods introduced. Authors provide comments only on boosting (in Section 3),  but many other methods have been presented. Just to provide few pointers to old works:

E. Alpaydin and C. Kaynak. Cascading classifiers.KYBERNETIKA, 34(4):369–374, 1998
L. Bruzzone and R. Cossu. A multiple-cascade-classifier system for a robust and par-tially unsupervised updating of land-cover maps. IEEE Transactions on Geoscienceand Remote Sensing, 40(9):1984–1996, 2002
Ludmila I. Kuncheva: Combining Pattern Classifiers - Methods and Algorithms , Wiley 2004

I agree with authors that the goal of their approach may be different (extracting shorter explanations rather than increasing classification performances), but a discussion of different cascading strategies may better contextualize and justify the proposed approach.

In the same spirit I would suggest the authors to discuss techniques for obtaining compact decision trees (pruning etc).


-> If we focus on classification accuracies, decision trees do not represent state-of-the-art classifiers, with performances which are very often far away from competitors like SVMs or Neural Networks. This somehow limits the application of these techniques (if not used inside Random Forests).


-> Authors provide methods and definitions (e.g. “valid explanation”) for a setting in which we have binary features. Why? If I correctly understand, the definitions can be easily provided also for numerical features (with the usual split based on a threshold), providing a more general framework. Moreover, the experiments are done with datasets having numerical non binary features.

---

### Official Review · AnonReviewer1 · 2020-10-31
**Recommending Rejection (Promising Work but Underdeveloped)**

**Rating:** 3
**Confidence:** 4

**Review:**


# Summary

This paper introduces a new type of classification model called the "cascading decision tree." The cascading decision tree is a rule-based classifier designed to have an overlapping hierarchical structure between its nodes to produce succinct explanations. The paper introduces these models, presents an induction algorithm to learn them from data, and includes an empirical evaluation on three UCI datasets as well as a propietary dataset. The submission includes code.

# Pros

1. The paper introduces a new kind of classification model. This model form is a contribution in and of itself (i.e., regardless of the algorithm used to fit cascading decision trees from data).

2. The paper highlights an innovative approach to the design of machine learning models – i.e., training models that are constrained to have particular "explainability" properties.

# Cons

3. The cascading trees produced by the algorithm in this work have little to no formal guarantees regarding their optimality or generalization produces. It is unclear if this is the best way to learn cascading decision trees.

4. The paper does not provide a pruning routine. The authors suggest that the algorithm can be paired with any generic pruning method. However, the empirical results do not showcase how the trees perform after pruning.

5. The experimental section is lacking in multiple ways. Ideally, this section should include comparisons on more than three datasets, and consider other baseline models such as "rule lists" (i.e., a special kind of decision tree) and sparse linear models (i.e., a type of model that does not require explanations). Finally, I would recommend the authors to include a plot that shows the distribution of explanation depths for all the examples in a dataset. This would allow readers to have a far better understanding of how each method affects the explanation depth (as compared to a comparison of the means).

6. The paper does not make a strong case to motivate why "shorter explanations are better." This is unfortunate given that succinct explanations are the primary motivation for using cascading decision trees. At a minimum, the paper should include a clear demonstration the advantages of using succinct explanations in a modern application. Ideally, this would include: (i) comparisons of the explanations produced for the same point by competing methods; (ii) a study of how the properties of explanations change based on other relevant phenomena (e.g., explanation depth for seen/unseen points).

# Rating

Overall, I was convinced that "cascading decision trees" were a valuable model class. I was also convinced that this work was valuable in that it highlights a novel approach for supervised learning (i.e., training models with explicit constraints on explainability such as in https://arxiv.org/abs/1703.03717)

My current rating (3) is based on the fact that the submission fails to analyze, validate, or motivate cascading decision trees sufficiently. Ideally, the paper should include a thorough analysis of the tree induction algorithm (as discussed in 3 and 4) as is standard in other work on decision trees. It should include more robust evidence that the proposed method produces succinct explanations (as discussed in 5), as well as a convincing demonstration of the utility of succinct explanations in modern applications (as discussed in 6).

# Questions

Q1. How does one measure the "quality of a cascading decision tree"? Is it only in terms of "explanation depth?"

Q2. Did you use a pruning routine in your experiments?

---

### Decision · Program_Chairs · 2021-01-07
**Final Decision**

**Decision:**

Reject

**Comment:**

This paper introduces the idea of cascading decision trees.  The reviewers agree that this is a potentially novel and valuable idea, but they also agree that the paper fall short in execution.  The paper would be substantially strengthened with more theoretical analysis, more discussion of why cascading decision trees are useful, and most importantly substantially more empirical evaluation, especially with more data sets and more baselines for comparison.